# ACCELERATED DEEP LEARNING BY GAUSSIAN CONTINUATION

## ABSTRACT

Prior work has shown that incorporating noise into the process of training deep neural networks reduces the risks of getting stuck in local minima, overfitting to the training data, and being limited by poor initialization. In this work we consider noisy training as a special case of optimization by continuation, also known as graduated non-convexity, where a convex version of the objective function is solved first and slowly morphed into the original non-convex function. When using continuation in machine learning problems, we show that saddle points require special consideration, as they may get the optimizer stuck in local minima. With a form of regularization applied to the continuation optimizer, we show on several test problems that this approach reduces the risk of being trapped in local minima, leading to better training for very deep architectures and non-convex loss functions.

## 1 INTRODUCTION

Training a deep neural network is known to be a non-convex optimization problem. The loss landscape is characterized by local minima and saddle points (Dauphin et al., 2014; Bottou et al., 2018), demanding a careful and generally empirical approach to choosing the model architecture and initialization. Model architectures that are highly flexible in principle may be useless in practice solely due to the difficulty of training them. An example of such would be a deep neural network of fully-connected layers applied to image classification.

How a model architecture affects the convexity of its loss function, and therefore its ease of training, has been studied extensively. A particular focus has been on the observation that deeper architectures, although more flexible, are also more difficult to train. Li et al. (2018) visualized this tradeoff on the loss landscape and, in line with much existing research (He et al., 2016; Tong et al., 2017; Orhan & Pitkow, 2018; Oyedotun et al., 2021), showed that skip connections are one way to promote convexity. Srivastava et al. (2015) investigated a variant of skip connections termed "information highways," inspired by long short-term memory networks. Sun et al. (2020), summarizing then-recent research, argues that "wide," overparameterized neural networks tend to have many equivalent local minima, making them easier to train by gradient-based methods.

Though deep networks present a considerable challenge, even shallow networks have suboptimal local minima that lead to challenges while training. Stochastic gradient descent (SGD) has however proven to be a powerful method in that it not only reduces the cost per iteration compared to full gradient descent, but the fact that the gradient is a noisy estimate leads to the ability to escape local minima (Bottou et al., 2018), especially "sharp" minima. Kleinberg et al. (2018) takes the view that SGD is in effect operating on a convolved (smoothed) version of the loss function, leading it to converge to "wider" minima. The nature of the noise that helps SGD reach better minima may not even be overly specific; if the noise class is general enough, SGD's performance and generalizability may be further improved by injecting noise artificially. Wu et al. (2020), Wei & Schwab (2019), Zhou et al. (2019), Neelakantan et al. (2017) and Orvieto et al. (2023) are a selection of works that investigate artificial noise in training, finding that it indeed improves SGD's ability to escape local minima and it acts as a form of regularization, improving generalizability. Ge et al. (2015) injects noise in order to improve SGD's ability to escape saddle points. Zhou et al. (2019), similar to Kleinberg et al. (2018), identify injected noise as effectively convolving the loss function with a kernel, smoothing it, and encouraging it to reach wider minima.

The concept of sharp vs. wide/flat minima refers to how sensitive the trained model is to perturbations in its parameters. It is generally observed and argued that wide minima generalize better (Orvieto et al., 2022; Chaudhari et al., 2019; Keskar et al., 2017), although this is not a strict correspondence in all cases (Dinh et al., 2017).

Promoting convexity in training may not only be done by modifying the model architecture or optimization algorithm, but also modifying the loss function. This is a well-established approach in computer vision (Blake & Zisserman, 1987; Terzopoulos, 1988; Yang et al., 2020), where it is known as graduated non-convexity. In machine learning, the loss function may be made more convex by modifying the dataset, such that the training dataset is presented with the easiest examples first and the hardest last, analogous to how a human might be taught; this approach is known as curriculum learning (Bengio et al., 2009). It has been successfully applied in reinforcement learning (Narvekar et al., 2020), and object localization, object detection, and machine translation (Soviany et al., 2022).

It is not only a useful interpretation of SGD's behaviour to say that it is optimizing a smoothed version of the loss function, this insight allows us to place noisy SGD in the framework of optimization by continuation. Continuation (Allgower & Georg, 2012) is an approach to minimization where a simplified objective is first defined whose solution is easier to obtain. The simplified objective is then gradually transformed into the original objective, while solving for intermediate solutions, until a solution to the original objective is obtained. It has been applied to optimization problems in computational chemistry (Moré & Wu, 1997; Wu, 1996) and computer vision (as mentioned above), and Bengio et al. (2009) identify continuation as the class of approach that includes curriculum learning. More recent work by Mobahi & Fisher III (2015b;a) provides a theoretical framework for continuation with Gaussian smoothing in general optimization problems. The convex envelope of a function was shown by Vese (1999) to be given by an evolutionary PDE with no analytical solution, and Mobahi & Fisher III (2015a) show that the best affine approximation to it is the heat equation. Since the solution to the heat equation is the Gaussian convolution of the non-convex function (Widder & Hirschman, 2015), Gaussian convolution is in this sense the optimal way to convexify a function. Gaussian continuation has since been applied in tensor PCA (Anandkumar et al., 2017), adversarial training (Iwakiri et al., 2022), and combinatorial optimization (Lin et al., 2023).

In our work, we approach the problem of training deep neural networks using continuation with Gaussian smoothing. Recent work by Iwakiri et al. (2022) forms the basis on which we build our approach, in particular their single-loop Gaussian homotopy method. We examine some of the practical difficulties in deep learning that do not satisfy key assumptions underpinning continuation methods, as well as how they might be addressed. We also show that despite the difficulties, continuation tends to reduce the variability in training performance and improves the training rate for deep architectures and non-convex loss functions. As with similar noisy optimization approaches, it tends to avoid local minima, especially sharp ones, improving generalization performance.

## 2 OPTIMIZATION BY CONTINUATION

Optimization by continuation is a technique for finding the minima of an objective function $f : \mathcal{M} \to \mathbb{R}$, where $\mathcal{M} \subset \mathbb{R}^m$. We consider an embedding of $f$ into a family of functions $g : \mathcal{M} \times \mathcal{L}$, where $\mathcal{L} = [0, \infty)$.

**Assumption 1.** *The family $g : \mathcal{M} \times \mathcal{L} \to \mathbb{R}$ has the following properties.*

1. *$g(\boldsymbol{\theta}, 0) = f(\boldsymbol{\theta})$,*

2. *$g(\boldsymbol{\theta}, \lambda)$ is bounded below,*

3. *$\lim_{\lambda \to \infty} g(\boldsymbol{\theta}, \lambda)$ is strictly convex in $\boldsymbol{\theta}$, and*

4. *$g(\boldsymbol{\theta}, \lambda)$ is continuously differentiable in $\boldsymbol{\theta}$ and $\lambda$.*

In the context of training a deep neural network, the objective function is a loss function $\ell$ evaluated over a finite dataset of feature-target pairs $\mathcal{D} = \left\{ (\boldsymbol{x}^1, \boldsymbol{y}^1), \dots, (\boldsymbol{x}^{n_d}, \boldsymbol{y}^{n_d}) \right\}$,

$$f(\boldsymbol{\theta}) = \frac{1}{n_d} \sum_{i=1}^{n_d} \ell(h(\boldsymbol{x}^i, \boldsymbol{\theta}), \boldsymbol{y}^i), \tag{1}$$

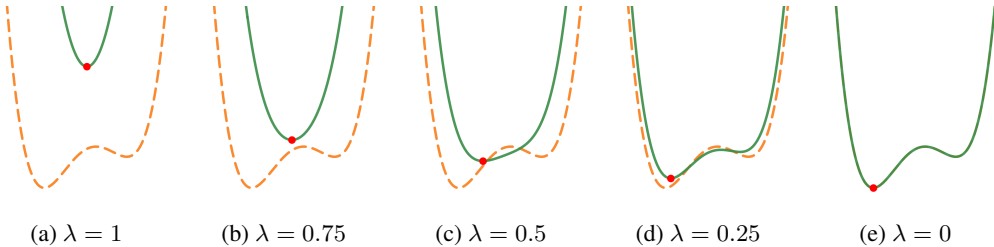

(a) $\lambda = 1$     (b) $\lambda = 0.75$     (c) $\lambda = 0.5$     (d) $\lambda = 0.25$     (e) $\lambda = 0$

Figure 1: Gaussian continuation applied to an non-convex objective function. The orange dashed line is the original objective $f(\theta)$, and the green solid line is the embedding $g(\theta, \lambda)$. The red dot is the intermediate minimum with respect to $\theta$.

where $h$ is our predictive model and $\boldsymbol{\theta}$ is the set of model parameters.

The continuation approach is summarized in Algorithm 1. We start with a sufficiently large $\lambda_0$ such that $g(\boldsymbol{\theta}, \lambda_0)$ is convex in $\boldsymbol{\theta}$, and find the corresponding minimizer $\boldsymbol{\theta}_0$. The continuation parameter is then decremented down to $\lambda_1$, and a new minimizer $\boldsymbol{\theta}_1$ is found numerically using $\boldsymbol{\theta}_0$ as the initial guess. This is repeated until $\lambda_n = 0$, where the corresponding $\boldsymbol{\theta}_n$ is a local minimum of the original objective function $f$. For this method to consistently reach the same minimum of $f$, we require the following additional assumption.

**Assumption 2.** *There exists a Lipschitz continuous curve through $\mathcal{M}$, parameterized by $\lambda$,*

$$\boldsymbol{\theta}^\star(\lambda) = \arg\min_{\boldsymbol{\theta}} g(\boldsymbol{\theta}, \lambda), \tag{2}$$

*which sweeps out a stationary path of $g$ originating at $\boldsymbol{\theta}_\infty^\star$, where $\boldsymbol{\theta}_\infty^\star = \lim_{\lambda \to \infty} \boldsymbol{\theta}^\star(\lambda)$. Per Assumption 1.3, $\lim_{\lambda \to \infty} g(\boldsymbol{\theta}, \lambda)$ is strictly convex in $\boldsymbol{\theta}$, therefore $\boldsymbol{\theta}_\infty^\star$ is unique.*

To the authors' knowledge, this assumption is made (either explicitly or implicitly) by most research into continuation methods (Mobahi & Fisher III, 2015a;b; Iwakiri et al., 2022; Lin et al., 2023). It means that, in principle, continuation turns a local optimization technique into a global one, in the sense that the final minimum it reaches is not sensitive to the initialization. This is because the continuation path always originates at the convex minimum $\boldsymbol{\theta}^\star(\lambda_0)$, and it is continuous from $\lambda_0$ to 0. This is visualized in Figure 1 for a univariate non-convex objective function.

Whereas $\boldsymbol{\theta}^\star(\lambda)$ is a continuous curve, hypothetically found by taking infinitesimal steps $\Delta\lambda$ in Algorithm 1, it can be shown that there exists some finite step size from $\lambda_i$ to $\lambda_{i+1}$ such that all intermediate minimizers lie along $\boldsymbol{\theta}^\star(\lambda)$.

**Theorem 1.** *If Assumption 2 holds, then there exists some $\Delta\lambda > 0$ such that, if the sequence $\lambda_0 > \cdots > \lambda_n$ provided as input to Algorithm 1 satisfies $\lambda_{i-1} - \lambda_i < \Delta\lambda \ \forall \ i \in \{1, \ldots, n\}$, the corresponding minimizers $\boldsymbol{\theta}_0, \ldots, \boldsymbol{\theta}_n$ lie along $\boldsymbol{\theta}^\star(\lambda)$.*

*Proof.* Let $\Delta\boldsymbol{\theta} \in \mathcal{M}$ with $\|\Delta\boldsymbol{\theta}\| < \epsilon$. Then Assumption 2 implies that there exists an $\epsilon > 0$ such that a gradient-based optimization algorithm applied to minimize $g$, with $\boldsymbol{\theta}^\star(\lambda) + \Delta\boldsymbol{\theta}$ as the initial guess, will always converge to the same minimizer $\boldsymbol{\theta}^\star(\lambda)$ for all $\lambda \geq 0$.

Assumption 2 also states that $\boldsymbol{\theta}^\star(\lambda)$ is Lipschitz continuous in $\lambda$, therefore the $\epsilon$ threshold on $\|\Delta\boldsymbol{\theta}\|$ has a corresponding $\Delta\lambda > 0$. In other words, $\Delta\lambda$ is the largest value that satisfies $\|\boldsymbol{\theta}^\star(\lambda + \Delta\lambda) - \boldsymbol{\theta}^\star(\lambda)\| < \epsilon \ \forall \ \lambda \geq 0$. $\qquad\square$

---

**Algorithm 1** Optimization by Continuation

---

   **input:** objective function $f : \mathcal{M} \to \mathbb{R}$, sequence $\lambda_0 > \cdots > \lambda_n = 0$
   $\boldsymbol{\theta}_0 = \arg\min_{\boldsymbol{\theta}} g(\boldsymbol{\theta}; \lambda_0)$.
   **for** $\lambda_i$ **in** $\lambda_1, \ldots, \lambda_n$ **do**
      $\boldsymbol{\theta}_i = \arg\min_{\boldsymbol{\theta}} g(\boldsymbol{\theta}; \lambda_i)$, initialized by $\boldsymbol{\theta}_{i-1}$
   **end for**
   **output:** $\boldsymbol{\theta}_n$

---

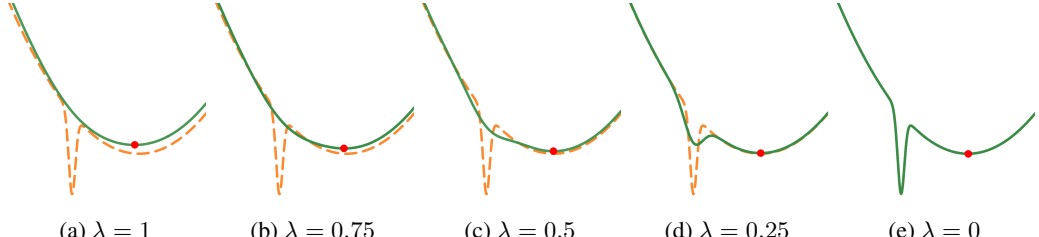

| (a) $\lambda = 1$ | (b) $\lambda = 0.75$ | (c) $\lambda = 0.5$ | (d) $\lambda = 0.25$ | (e) $\lambda = 0$ |

Figure 2: A scenario where Gaussian continuation does *not* reach the global minimum.

## 2.1 Gaussian Convolution Embedding

The approach taken by Mobahi & Fisher III (2015b;a) and Iwakiri et al. (2022) is to take the convolution of $f$ with a Gaussian kernel $k(\boldsymbol{\theta}; \boldsymbol{\mu}, \boldsymbol{\Sigma})$ with mean $\boldsymbol{\mu} = \mathbf{0}$ and covariance $\boldsymbol{\Sigma} = \lambda \boldsymbol{I}$, which we denote by $k_\lambda(\boldsymbol{\theta})$,

$$g(\boldsymbol{\theta}; \lambda) = [f \star k_\lambda](\boldsymbol{\theta}) = \int_{\mathcal{M}} f(\boldsymbol{\vartheta}) k_\lambda(\boldsymbol{\theta} - \boldsymbol{\vartheta}) d\boldsymbol{\vartheta}, \tag{3}$$

which is also known as the *Weierstrass transform*. This choice of $g$ constrains the permissible class of $f$ as described in Section 3 of Zemanian (1967). For all permissible $f$, it is a unique and invertible transform (Widder & Hirschman, 2015; Shapiro, 1966). The kernel $k_\lambda$ approaches a Dirac delta function as $\lambda \to 0$, therefore $g$ approaches $f$. The transform can also be interpreted as a solution to the heat equation, with $f(\boldsymbol{\theta})$ denoting the initial condition and $\lambda$ interpreted as time (Widder & Hirschman, 2015).

Although Assumptions 1.1, 1.2, and 1.4 are guaranteed for our definition of $g$, 1.3 imposes the following condition that our objective function must satisfy:

$$\lim_{\lambda \to \infty} \frac{\partial^2 g}{\partial \boldsymbol{\theta}^2}(\boldsymbol{\theta}; \lambda) = \lim_{\lambda \to \infty} \left[ \frac{d^2 f}{d\boldsymbol{\theta}^2} \star k_\lambda \right](\boldsymbol{\theta}) \text{ is positive definite.} \tag{4}$$

In other words, as $f$ is smoothed out by increasing $\lambda$, it must approach a convex function. This is not guaranteed in general by a loss function like equation 1, however this can be fixed by including a regularization term.

Although the minimum reached by continuation is not guaranteed to be the global minimum of $f$, as long as Assumptions 1 and 2 hold, it is guaranteed that no matter the initialization, optimization by continuation should reach the same minimum. A scenario where the minimum reached by continuation is *not* the global one is shown in Figure 2. In this example, there are two minima, and although the global minimum is on the left, continuation reaches the local minimum on the right. This is because that minimum is more robust to the Gaussian smoothing of the objective. In this way, continuation ignores sharp minima of $f$, which tends to improve generalization.

A connection to variational inference may also be made. If the objective function $f$ is a log-likelihood function, then optimizing it by Gaussian continuation is equivalent to a form of variational inference whereby the variational distribution is a fixed-covariance Gaussian. This is explained in detail in Appendix A.

## 2.2 Adapting the Continuation Parameter

This section discusses different strategies for adapting $\lambda$ during optimization. The example function from Figure 1 is shown as a 3D plot over both $\theta$ and $\lambda$ in Figure 3.

A straightforward implementation of Algorithm 1 suggests an optimization trajectory resembling that in Figure 3a, in that we solve a separate minimization problem in $\boldsymbol{\theta}$ for each decrement of $\lambda$, and $\lambda$ is adapted by its own scheme. An example of such a method is the approach used by Neelakantan et al. (2017), which is a fixed variance decay of the form $\lambda_i = \lambda_0/(1 + i)^\gamma$, where $\lambda_0$ is the initial value of $\lambda$, $\gamma$ is a hyperparameter, and $i$ is the iteration number (as in Algorithm 1). Another example is that used by Zhou et al. (2019), which is a geometric annealing schedule, $\lambda_i = \lambda_0 \gamma^i$, where $0 < \gamma < 1$ is again a hyperparameter.

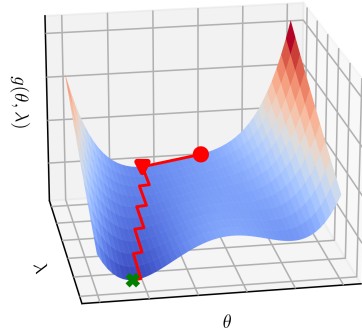
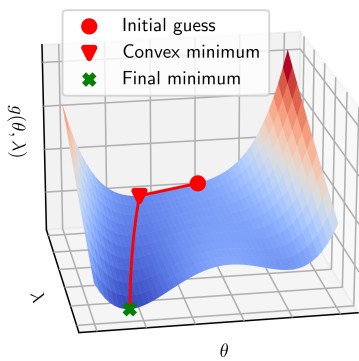

(a) $\lambda$ decrementing separate from optimization

(b) $\lambda$ as an optimization variable

Figure 3: Optimization of example from Figure 1 with different schemes for adapting $\lambda$

---

**Algorithm 2** Optimization by Continuation with $\lambda$ as Optimization Variable

---

**input:** objective function $f : \mathcal{M} \to \mathbb{R}$, initial continuation parameter $\lambda_0$, number of iterations $n$
$\boldsymbol{\theta}_0 = \arg\min_{\boldsymbol{\theta}} g(\boldsymbol{\theta}; \lambda_0)$.
**for** $i$ **in** $1, \ldots, n$ **do**
  Calculate $\left(\frac{\partial g}{\partial \boldsymbol{\theta}}\right)_{i-1}$ and $\left(\frac{\partial g}{\partial \lambda}\right)_{i-1}$.
  Take optimization step, e.g. by gradient descent with learning rates $a$, $b$:
$$\boldsymbol{\theta}_i = \boldsymbol{\theta}_{i-1} - a\left(\frac{\partial g}{\partial \boldsymbol{\theta}}\right)_{i-1} \qquad \text{and} \qquad \lambda_i = \max\left(0, \lambda_{i-1} - b\left(\frac{\partial g}{\partial \lambda}\right)_{i-1}\right)$$
**end for**
**output:** $\boldsymbol{\theta}_n$

---

An alternative approach proposed by Iwakiri et al. (2022) is to treat $\lambda$ as simply another optimization variable, to be adapted alongside $\boldsymbol{\theta}$ in order to minimize $g$. This approach is shown in Algorithm 2. However, since we are seeking a minimum of $f(\boldsymbol{\theta})$, the validity of this approach depends on whether $\lambda$ naturally tends toward zero as $g(\boldsymbol{\theta}, \lambda)$ is minimized.

**Theorem 2.** *The minimized objective $g^\star(\lambda) = g(\boldsymbol{\theta}^\star(\lambda), \lambda)$ increases monotonically with $\lambda$, i.e.,*

$$g^\star(\lambda_1) < g^\star(\lambda_2) \quad \text{if and only if} \quad \lambda_1 < \lambda_2. \tag{5}$$

*Proof.* See Appendix B for full proof. To summarize, we show that the following identity holds,

$$\text{tr}\left(\frac{\partial^2 g}{\partial \boldsymbol{\theta}^2}\right) = 2\frac{\partial g}{\partial \lambda}, \tag{6}$$

therefore whenever the trace of the Hessian of $g$ with respect to $\boldsymbol{\theta}$ is positive (as is the case at the minimum $\boldsymbol{\theta}^\star(\lambda)$), the derivative of $g$ with respect to $\lambda$ is positive, therefore $g$ is monotonic in $\lambda$. $\square$

This was also demonstrated by Iwakiri et al. (2022) using the fact that the Gaussian convolution is the solution to the heat equation. This produces an optimization path resembling that in Figure 3b, eliminating the need for a specialized $\lambda$ adaptation scheme.

Analogous to Theorem 1, under certain circumstances, optimizing by Algorithm 2 approximately follows the curve $\boldsymbol{\theta}^\star(\lambda)$.

**Theorem 3.** *Using Algorithm 2, with sufficiently small step size in $\lambda$ relative to $\boldsymbol{\theta}$, the sequence $\boldsymbol{\theta}_0, \ldots, \boldsymbol{\theta}_n$ approximates $\boldsymbol{\theta}^\star(\lambda_0), \ldots, \boldsymbol{\theta}^\star(\lambda_n)$.*

*Proof.* By Theorem 1, we know that there is some radius $\epsilon$ around $\boldsymbol{\theta}^\star(\lambda)$ such that for any initial guess $\boldsymbol{\theta}^\star(\lambda) + \Delta\boldsymbol{\theta}$ where $\|\Delta\boldsymbol{\theta}\| < \epsilon$, a gradient-based optimizer will converge to $\boldsymbol{\theta}^\star(\lambda)$. Therefore, if the step size for $\lambda$ in Algorithm 2 corresponds to a perturbation in $\boldsymbol{\theta}$ that is less than $\epsilon$ from $\boldsymbol{\theta}^\star(\lambda)$, the optimizer should still tend toward $\boldsymbol{\theta}^\star(\lambda)$. $\square$

## 2.3 MONTE CARLO CONTINUATION

In training a deep neural network, $\mathcal{M}$ may be a high-dimensional ($m > 10^5$) vector space, which makes the convolution prohibitively difficult to evaluate by quadrature methods. In addition, evaluating $f(\boldsymbol{\theta})$ by equation 1 in practice does not involve the whole dataset, but a minibatch $\mathcal{B}$. The Monte Carlo approximation of the convolved objective takes a doubly stochastic form,

$$[f \star k_\lambda](\boldsymbol{\theta}) \approx \frac{1}{N|\mathcal{B}|} \sum_{j=1}^{N} \sum_{i \in \mathcal{B}} \ell(h(\boldsymbol{x}^i, \boldsymbol{\vartheta}_j), \boldsymbol{y}^i), \tag{7}$$

where $\{\boldsymbol{\vartheta}_1, \ldots, \boldsymbol{\vartheta}_N\} \sim \mathcal{N}(\boldsymbol{\theta}, \lambda \boldsymbol{I})$. In the general case, this means that whereas standard optimization of $f$ requires one function evaluation per step of the optimizer, optimization by Monte Carlo continuation requires $N$ evaluations. If evaluating $f$ dominates the cost of optimization, this may make continuation prohibitively expensive. For this reason, the special case of $N = 1$ is considered for the large learning problems in Section 3. This means that Monte Carlo continuation is equivalent to adding Gaussian noise to model parameters before each optimizer step, the additional cost of which is likely negligible.

An unbiased, doubly stochastic approximation of the gradient with respect to $\lambda$ may be written as

$$\begin{aligned}
\frac{\partial}{\partial \lambda}[f \star k_\lambda](\boldsymbol{\theta}) &= \int_{\mathcal{M}} f(\boldsymbol{\vartheta}) \frac{\partial k_\lambda}{\partial \lambda}(\boldsymbol{\theta} - \boldsymbol{\vartheta}) d\boldsymbol{\vartheta} \\
&\approx \frac{1}{N|\mathcal{B}|} \sum_{j=1}^{N} \sum_{i \in \mathcal{B}} \frac{(\boldsymbol{\theta} - \boldsymbol{\vartheta}_j)^T (\boldsymbol{\theta} - \boldsymbol{\vartheta}_j) - m\lambda}{2\lambda^2} \ell(h(\boldsymbol{x}^i, \boldsymbol{\vartheta}_j), \boldsymbol{y}^i),
\end{aligned} \tag{8}$$

where $\{\boldsymbol{\vartheta}_1, \ldots, \boldsymbol{\vartheta}_N\} \sim \mathcal{N}(\boldsymbol{\theta}, \lambda \boldsymbol{I})$. It is worth noting that evaluating this gradient does not necessarily require any extra function evaluations, as the samples of $f(\boldsymbol{\vartheta}_i)$ may be reused from equation 7.

Using the analysis presented by Iwakiri et al. (2022) along with appropriate assumptions (see Theorem 3.5), it can be shown that the number of iterations required to find $(\hat{\boldsymbol{\theta}}, \hat{\lambda})$ that satisfy $\|\nabla g(\hat{\boldsymbol{\theta}}, \hat{\lambda})\|_2 < \varepsilon$ scales as $\mathcal{O}(1/\varepsilon^4)$, which matches the iteration complexity of standard stochastic gradient descent (Ghadimi & Lan, 2013). This holds under the assumption that $\lambda_i \leq \lambda_0 \gamma^i$, where $\gamma$ is a hyperparameter.

## 2.4 SADDLE POINTS IN OBJECTIVE FUNCTIONS

Assumption 2 implies that symmetric saddle points may not exist along the continuation path $\boldsymbol{\theta}^\star(\lambda)$, because if they do, it would imply the existence of a bifurcation in $\boldsymbol{\theta}^\star(\lambda)$. These saddle points however are known to be present in machine learning problems (Dauphin et al., 2014). We illustrate the impact of these on optimization behaviour through an $m = 2$-dimensional test function,

$$f(\boldsymbol{\theta}) = a\left(\delta_1^2 + \delta_2^2\right) - \exp\left(-\frac{\delta_1^2}{2}\right) - \exp\left(-\frac{\delta_2^2}{2}\right), \quad \delta_1^2 = \|\boldsymbol{\theta} - \hat{\boldsymbol{\theta}}_1\|^2, \ \delta_2^2 = \|\boldsymbol{\theta} - \hat{\boldsymbol{\theta}}_2\|^2, \tag{9}$$

which has two equivalent minima near $\hat{\boldsymbol{\theta}}_1$ and $\hat{\boldsymbol{\theta}}_2$ (see Table 1 in Appendix E). We set them to $\hat{\boldsymbol{\theta}}_1 = (2, \ldots, 2)$ and $\hat{\boldsymbol{\theta}}_2 = (-2, \ldots, -2)$, putting the saddle point at the origin. The remaining hyperparameter is $a$, for which we consider two values $a \in \{0.01, 0.02\}$. We use simple gradient descent with $\lambda_0 = 3$. The optimizer is run for a "warmup period," meaning $\lambda$ is held constant at $\lambda_0$ for a certain number of iterations. The optimizer converges on the convex minimum during the warmup period, after which $\lambda$ is also adapted by gradient descent. The learning rates for $\boldsymbol{\theta}$ and $\lambda$, the warmup period, and the total number of iterations are given in Appendix E.

There are two ways in which the bifurcation in $\boldsymbol{\theta}^\star(\lambda)$ can impact optimization behaviour. Figure 4 shows both scenarios.

For the case where $a = 0.01$, shown in Figure 4b, the Hessian trace stays positive after the bifurcation occurs. What this means is that, after the origin transitions from the convex minimum to a saddle point, $\frac{\partial g}{\partial \lambda}$ is no longer guaranteed to be positive, and so the value of $\lambda$ stops decreasing. However, since the origin is already a saddle point at this intermediate value of $\lambda$, the intermediate

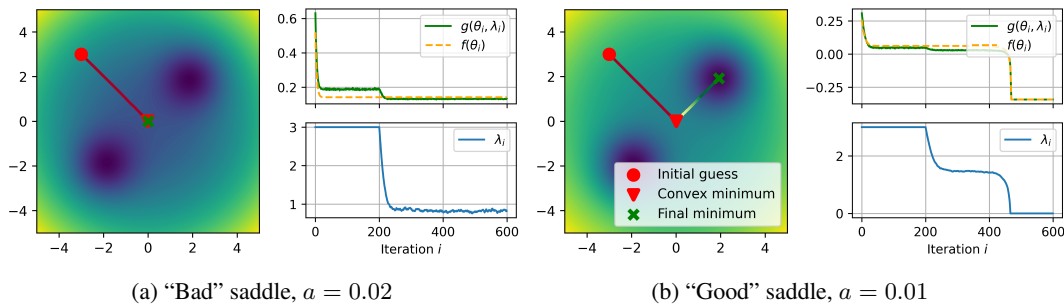

(a) "Bad" saddle, $a = 0.02$          (b) "Good" saddle, $a = 0.01$

Figure 4: Optimization path on both variants of the saddle function. In the loss landscape plots, the colour of the trace corresponds to the value of $\lambda$.

minimum is allowed to drift toward either true minimum. As the optimizer commits to one of the true minima, $\lambda$ decreases again until it reaches zero. The only difference between this scenario and one that satisfies Assumption 2 is that the final minimum is no longer deterministic. Because it still reaches a minimum however, we refer to this as the "good" saddle.

For the case where $a = 0.02$, shown in Figure 4a, the Hessian trace turns negative before the bifurcation occurs. Again $\lambda$ stagnates at a nonzero value, but this corresponds to a $g$ that still has a convex minimum at the origin. If we simply follow Algorithm 2, we are now stuck, as $g$ is minimized with respect to both $\boldsymbol{\theta}$ and $\lambda$. We refer to this as the "bad" saddle. This situation may only be escaped by either forcing $\lambda$ to decrease artificially, or by using few Monte Carlo samples, which would encourage the optimizer to escape the saddle point. The hyperparameter values for the optimization tests are summarized in Table 2 in Appendix E.

## 2.5 REGULARIZING THE $\lambda$ GRADIENT ESTIMATOR

The goal of this regularization is to bias the gradient estimator in equation 8 toward positive values, decreasing $\lambda$ during optimization. There are two reasons to do this. The first is to counteract the saddle point problem described above, as a bad saddle may stall the optimization. The second is, as mentioned in Section 2.3, the estimators are used with one Monte Carlo sample in order to mitigate computational cost, so the estimator is likely to have high variance in practice. By random chance, this may lead to excessive growth in $\lambda$, compromising the training.

Our regularization strategy for $g$ is inspired by $\ell_2$ regularization of $\boldsymbol{\theta}$. Consider embedding the regularized loss $f_r(\boldsymbol{\theta}; \beta) = f(\boldsymbol{\theta}) + \beta\boldsymbol{\theta}^T\boldsymbol{\theta}$,

$$g(\boldsymbol{\theta}; \lambda, \beta) = [f_r \star k_\lambda](\boldsymbol{\theta}) = [f \star k_\lambda](\boldsymbol{\theta}) + \left[(\beta\boldsymbol{\theta}^T\boldsymbol{\theta}) \star k_\lambda\right](\boldsymbol{\theta}), \tag{10}$$

where $\beta$ is a new regularization hyperparameter. The corresponding Hessian is

$$\frac{\partial^2 g}{\partial\boldsymbol{\theta}^2} = \left[f \star \frac{\partial^2 k_\lambda}{\partial\boldsymbol{\theta}^2}\right](\boldsymbol{\theta}) + 2\beta\boldsymbol{I}. \tag{11}$$

As mentioned in Theorem 2, a general fact of the Gaussian convolution is that $\frac{1}{2}\mathrm{tr}\left(\frac{\partial^2 g}{\partial\boldsymbol{\theta}^2}\right) = \frac{\partial g}{\partial\lambda}$, so the regularized $\lambda$ gradient estimator becomes

$$\frac{\partial g}{\partial\lambda} = \frac{1}{2}\left[f \star \mathrm{tr}\left(\frac{\partial^2 k_\lambda}{\partial\boldsymbol{\theta}^2}\right)\right](\boldsymbol{\theta}) + \beta m = \left[f \star \frac{\partial k_\lambda}{\partial\lambda}\right](\boldsymbol{\theta}) + \beta m. \tag{12}$$

Because $\left[f \star \frac{\partial k_\lambda}{\partial\lambda}\right](\boldsymbol{\theta})$ is the gradient estimator from equation 8 with unregularized loss, the regularization simply consists of adding a constant $\beta m$ to $\frac{\partial g}{\partial\lambda}$. For actual implementation, a change of variables was introduced so that $\lambda$ would not need explicit bounds; this is discussed in Appendix C.

## 3 NUMERICAL STUDIES

The Monte Carlo continuation approach is assessed using different kinds of test problems. In the body of the paper, two are presented: a deep neural network applied to image classification, and deep

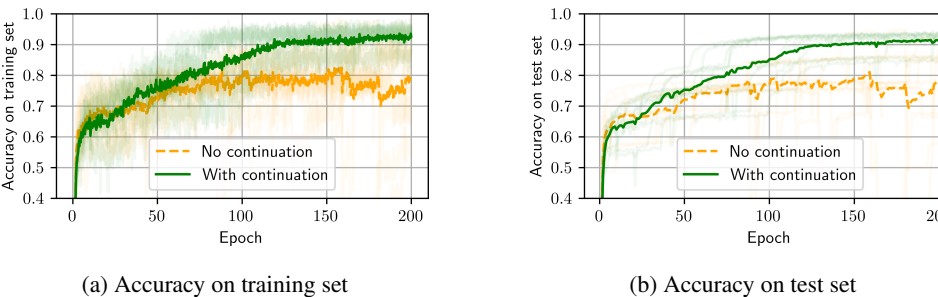

(a) Accuracy on training set        (b) Accuracy on test set

Figure 5: MNIST classifier: mean convergence after training an ensemble of 10 models with continuation and 10 without.

neural ODEs applied to learning parameterized dynamics equations from a time series dataset. For both of these, due to the cost of evaluating the loss function and gradient, continuation is performed with only one Monte Carlo sample. In other words, continuation amounts to adding Gaussian noise to the model parameters during training. In Appendix D, continuation is applied to non-convex 2D test functions commonly used to assess optimization methods. The optimization setup and visualization are similar to Section 2.4.

### 3.1 CLASSIFICATION NEURAL NETWORK

This test problem is inspired by an example from Neelakantan et al. (2017): a deep classification network applied to the MNIST handwritten digit dataset (LeCun et al., 1998). The neural network has 30 fully-connected layers, and each hidden layer has 50 units. The ReLU activation (Nair & Hinton, 2010) is applied after each hidden layer, and a softmax is applied to the output layer. Dense networks such as this are usually avoided in image classification in favour of convolutional networks due to large number of trainable parameters, and consequently, the difficulty of training them. It therefore makes a useful test for continuation's ability to overcome architecture non-convexity. The Adam optimizer (Kingma & Ba, 2015) is used with a learning rate of $10^{-4}$ to train for 200 epochs. A learning rate of $10^{-2}$ is used for the continuation parameter, and a regularization weight of $10^{-3}$ for the gradient estimate. The initial continuation parameter is $\lambda_0 = \exp(-13)$ (see the change of variables in Appendix C).

The results of training are shown in Figure 5. For each scenario (with or without continuation), the experiment consisted of training an ensemble of 10 models. This is because the performance varies significantly from run to run, so we compare the mean over the ensemble. The models trained without continuation tend to stagnate at accuracy levels between 60% and 90%, indicating that they are trapped in local minima, whereas the models trained with continuation tend to escape these local minima and reach the highest accuracy stratum of $\sim 94\%$. This not only achieves greater accuracy, but reduces variability in training.

### 3.2 LEARNING DYNAMICS WITH NEURAL ODES

The final test is to learn a system of parameterized ODEs based on a time series dataset, using a neural ODE. Two dynamical systems are considered here: the Lotka-Volterra system of predator and prey populations, and the Lorenz system. The details of the setup are given in Appendix F. For each learning problem, the dataset consists of 30 different parameter instances and 30 corresponding time series, which are solutions to the system at evenly-spaced time steps. To divide both datasets into train/test sets, we use the same approach as with the MNIST classifier, obtaining the test set by randomly selecting 10% (three) of the parameter instances and their corresponding time series. No time series minibatching is used in either problem; the neural ODE model takes the parameters as input and outputs the full time series.

The neural ODEs used here have a similar architecture to the classification network above. Each model consists of simple fully-connected layers, and each hidden layer again has 50 units. The Lotka-Volterra model uses only 5 layers, whereas the Lorenz model is much deeper at 20 layers.

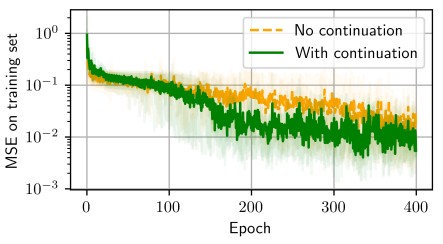
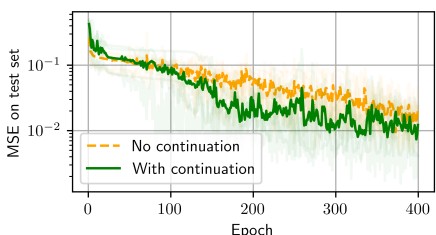

(a) Mean squared error on training set

(b) Mean squared error on test set

Figure 6: Lotka-Volterra neural ODE: mean convergence after training an ensemble of 5 models with continuation and 5 without.

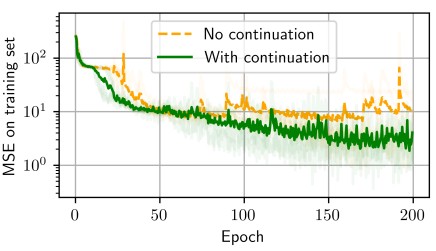
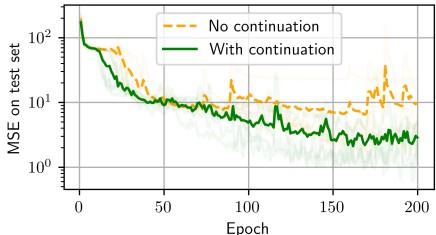

(a) Mean squared error on training set

(b) Mean squared error on test set

Figure 7: Lorenz neural ODE: mean convergence after training an ensemble of 5 models with continuation and 5 without.

The ReLU activation is applied after each hidden layer, and there is no activation on the output layer. The Adam optimizer is used for both models. For the Lotka-Volterra model, the model parameter learning rate is $10^{-3}$, the continuation parameter learning rate is $3 \times 10^{-1}$, and the gradient regularization weight is $10^{-1}$. It is trained for 400 epochs. For the Lorenz model, the model parameter learning rate is $3 \times 10^{-3}$, the continuation parameter learning rate is $3 \times 10^{-1}$, and the gradient regularization weight is $10^{-2}$. It is trained for 200 epochs. For both problems, the initial continuation parameter is $\lambda_0 = \exp(-10)$ (see the change of variables in Appendix C).

The results of training are shown in Figures 6 and 7. Similar to the MNIST classifier, the experiment consisted of training an ensemble of 5 models with continuation and 5 without. With Lotka-Volterra, the models trained without continuation are not necessarily stuck in a local minimum, however continuation still speeds up the descent toward better minima. With Lorenz, the models trained without continuation tend to converge on a mean squared error (MSE) of roughly $10^1$, indicating that they are trapped in local minima, but there are also sharp increases in MSE during training, suggesting the minimum reached is highly sensitive to optimization steps. The models trained with continuation tend to escape the local minima and reach lower MSE, and tend to be more stable.

## 4 CONCLUSION

We have discussed the application of Gaussian continuation methods to deep learning problems, and in particular how optimizing the continuation parameter compares to the classical approach of a fixed schedule. We show that, if the continuation path exists, finite implementations of both approaches are able to follow it. We also show that, although saddle points may stall the optimization process, this may be addressed with a simple regularization term added to the continuation gradient estimator. From a theoretical viewpoint, the convergence rate of Gaussian continuation matches that of standard SGD under appropriate assumptions. The numerical studies presented in this paper suggest that in practice, Gaussian continuation, applied to any standard gradient-based optimizer, consistently converges faster to better minima. In addition, optimization with Gaussian continuation is less sensitive to initialization, reducing variability in training.

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
