# OpenReview forum: "Accelerated Deep Learning by Gaussian Continuation"
_ICLR.cc/2024/Conference — ICLR 2024 Conference Withdrawn Submission_

### Official Review · Reviewer_EAjK · 2023-10-28

**Soundness:** 2 fair
**Presentation:** 2 fair
**Contribution:** 3 good
**Rating:** 5
**Confidence:** 3

**Summary:**

The paper presents an analysis of optimization by continuation and relates it to the addition of noise during training and the avoidance of sub-optimal local minima.

**Strengths:**

I believe that the combination of (i) optimization by continuation, (ii) its Monte Carlo approximation, (iii) use of regularization to improve its numerical solution, and (iii) the idea of adding noise to avoid local minima; is of importance. The paper has done a good effort in trying to explore all these points together to deliver its contribution. The application to Machine Learning is clear from the numerical experiments.

**Weaknesses:**

I found the paper needs to be greatly improved in its writing, especially when presenting its mathematical derivations and its contributions. The presentation of ideas needs to be improved likewise; many (sub)sections are presented without a cohesive narrative that let the reader understand it. I now present my concerns.

-> The main idea of the paper is to explain how adding noise during training helps avoiding local minima by making a connection with optimization by continuation. Such connection depends on the Monte Carlo approximation of the continuation, so that, when one sample is used, it somehow becomes equivalent to adding noise during training. I believe such connection is an important one in the paper, but I do not see it explicitly stated in the introduction – it took me reading section 2.4 and the first paragraph of section 3 to realize this is the case! This doesn’t help the common reader – more emphasis has to be placed on the writing of the paper to emphasize such connection right from the beginning. (Unless I am missing some details). Now, the other tricky thing is that the paper also updates the continuation parameter, and so it does not simply add noise to updates like other papers do. This is a difference that also should be highlighted.

-> So, is this paper the first one using continuation + reguilarization + MC approximation to find better minima for optimization? Previous paper have added noise to training, but they do not seem to also use continuation in the same way the paper does. Is that right? If previous works have used continuation in optimization, how is this paper different from the rest? I am trying to understand that, since there are previous works about continuation like Iwakiri et al. (2022), Neelakantan et al. (2017), and Zhou et al. (2019).

-> As mentioned, the paper uses regularization and justifies its inclusion during training. Do previous works that add noise to the training steps also add some kind of regularization? This is unclear to me.

-> I also think that a third curve to the simulations should be added, where continuity is used without regularization!! This is a very interesting setting, since it may be what other people has done in the past and would serve as a better motivation for the inclusion of regularization. We need to see its performance.

-> Why does the paper have “ACCELERATED deep learning” in the title? I see no argument for acceleration in the paper. I can see that better minima might be achieved in the experiments, but I don’t think that implies “acceleration” – I would expect a more formal or some other comparisons that can show such “acceleration” during training.

==

-> I also think the contributions of the paper must be listed, since they are all condensed in the last paragraph of the introduction. Also, add response to the following questions for strengthening the contribution: when studying continuation in deep learning, why do we need to care about it? What are the repercussions of the results in the paper?

==

-> In Algorithm 1, nothing states what kind of update there is for the parameters. You have to read the paper to infer some gradient descent method is being used (e.g., before assumption 2) – but this is not a good thing, because the algorithm 1 should be self-contained! You also need to specify the initialization at time zero.
-> Another issue I find regarding algorithm 1 is that descent methods may ensure convergence asymptotically, thus, the minima cannot be achieved in finite-time. However, this seems to contradict the fourth line of Algorithm 1 and its analysis. Please, correct this.

-> Why do we care about the different schemes for adapting $\lambda$ presented throughout the paper before Algorithm 2? Section 2.2 is introduced without much explanation.

-> There is a grave confusion in Algorithm 2 and the proof of Theorem 2: it turns out that there is no explicit mention over which values the derivative of g with respect to $\theta$ and $\lambda$ are evaluated on. This makes their derivatives indefinite and all the derivations that use them hard to follow.

-> Theorem 3 is stated in a confusing way: wouldn’t it be better to say “for a sufficiently small $a/b$”?

-> In the paragraph before subsection 2.4, it talks about iteration complexity being similar to SGD. Does it refer to Algorithm 2 or something with Monte Carlo continuation? This paragraph seems a little bit unrelated to the rest of subsection 2.3 – I think the writing has to be improved to know its connection with the rest of the subsection.

-> Subsection 2.4 talks about bifurcation, what does it mean? Also, provide more explanation about how the signs of the eigenvalues affect finding a minima as related to changes in the optimization landscape and the bifurcation phenomenon.

-> I am curious about a question: why do the training and test error of the Neural Lorenz ODE seem to increase as epochs continue? Is it some sort of instability?

==

-> I suggest adding a brief intro to Neural ODEs in the appendix. I think most readers are less familiar with them compared to traditional neural networks for classification, for example.

**Questions:**

Please, see the questions made on the "weaknesses" section of the review.

---

### Official Review · Reviewer_MQoK · 2023-10-30

**Soundness:** 1 poor
**Presentation:** 3 good
**Contribution:** 2 fair
**Rating:** 3
**Confidence:** 3

**Summary:**

Gaussian continuation is the approach of convexifying and smoothing a given loss function by convolution with a Gaussian kernel. The variance/width of the kernel is controlled by an additional parameter lambda which is chosen at a larger value earlier in the optimization and then decayed to 0 yielding the original loss function in the end. The goal is to optimize a convexified version of the loss early in training to avoid local minima and get close to more close to global minima. In this paper the authors propose to apply a Gaussian continuation approach to deep learning problems by estimating the convolution with a Gaussian via a Monte Carlo estimator of the convolution integral by drawing a single sample so as to stay computationally feasible. The authors describe how Gaussian continuation and their Monte Carlo estimate can behave around a simplistic 2D saddle point problem and how a regularized version of the Monte Carlo estimated convolved loss can be used to better estimate the derivative with respect to lambda to avoid problems due to high variance of the single sample Monte Carlo estimate. The authors show results from numerical experiments to demonstrate the practical usefulness of their proposed approach arguing that it outperforms a baseline that does not leverage Gaussian continuation on MNIST classification and neural ODE training.

**Strengths:**

- The paper gives a good introduction, formal treatment and extensive background of the basic concepts and background literature on Gaussian continuation

**Weaknesses:**

- The numerical experiments seem severly flawed in multiple ways. The baseline MNIST classifier without continuation is nowhere near what a simple fully connected 2-layer neural network trained with SGD can achieve on MNIST in terms of test set accuracy (their baseline training set accuracy is below 80% while it is easy to get above 90% test accuracy and should be easy to memorize the training set on MNIST). Furthermore, the variance across different training runs the authors show in their figures, especially for the baseline (without continuation) looks very alarming. Their baseline training setup is too unstable to average across and be considered anything remotely close to a fair baseline.
- Given the minimal numerical results it does not become clear whether the added dimension of how to decay lambda properly is really given a robust solution with the approach described in the paper

**Questions:**

- If the focus is on high dimensional deep learning problems perhaps do not use so much space on 1D intuitions (e.g. Figure 1 and Figure 2). It is also not clear to me that the saddle point example in 2.4 is very meaningful to how saddle points in the loss landscape for high dimensional neural networks look like. I would instead recommend to focus more on how Gaussian continuation interplays with the noise from SGD (different noise distribution than normal Gaussian) and its well known annealing affects typically leveraged through learning rate schedules. Does injecting noise with the different distribution really help and if yes in what circumstance in a high dimensional loss landscape? Some papers in the direction of this question (like https://proceedings.mlr.press/v162/orvieto22a.html) are already cited in the paper.
- Nit: In the abstract it says saddle points are responsible for getting stuck in local minima, but a saddle point is not a minimum, just a zero gradient point to get stuck in?

---

### Official Review · Reviewer_xWFa · 2023-10-30

**Soundness:** 3 good
**Presentation:** 1 poor
**Contribution:** 1 poor
**Rating:** 3
**Confidence:** 4

**Summary:**

In this work, the authors consider Gaussian continuation to solve smooth optimization problems.
The authors show that sampling can be useful for solving optimization problems.

**Strengths:**

The authors show that the Gaussian convolution is useful for optimization problem.
Under certain assumptions, the authors give some theoretical results.
Some toy experiments are provided to support the claim.

**Weaknesses:**

My main question is what is the main contribution of this work.
The Gaussian formulation has been considered in evolution strategies [1,6,13,14,19],  variational optimization [17,18], and variational inference [2,3,8,9].
The authors do not mention prior works in these related fields.
It is essential to know these prior works in order to properly gauge the contributions of this work.
Many theoretical results such as convergence analysis and gradient estimation in this paper are known in these fields.

Numerical results in this work are also very weak. Prior work [17] considered a similar problem for neural networks and showed strong numerical results.
The authors of this work claim that the proposed method is useful for very deep architectures while they only consider MLPs in some toy datasets such as MNIST.  Empirically, it has been shown that sampling from a Gaussian can be useful for generalisation [16,17].  Thus, the claim that continuation/sampling is useful for  optimization in deep learning is not new.

I fail to see the main contribution of this work given that many theoretical and empirical results have been considered in the related fields.
The authors should also discuss the difference between Gaussian continuation and evolution strategies including  https://en.wikipedia.org/wiki/Gaussian_adaptation. More related works about Gaussian adaption can be found on the Wikipedia page.

The following points have been considered in the literature while the authors fail to include related works.

* In Eq (3), the authors consider the following (equivalent)  Gaussian convolution.
$
\ell(\theta,\Sigma) = E_{ z \sim q(z; \theta,\Sigma)} [ f(z) ] + \gamma  E_{ z \sim q(z; \theta,\Sigma)} [ \log q(z; \theta,\Sigma) ],
$ where $q(z; \theta,\Sigma)$ is a Gaussian approximation with mean $\theta$ and covariance $\Sigma$.
In evolution strategies [1,6,7], a similar formulation has been considered with $\gamma =0$ and non-differentiable $f(z)$ using a full or diagonal Gaussian approximation. In variational optimization[17,18],  a similar formulation has been considered with $\gamma =0$ and differentiable $f(z)$ using a full or diagonal Gaussian approximation. In variational inference [2,8], another similar formulation has been considered with $\gamma =1$ and differentiable $f(z)$ using a full or diagonal Gaussian approximation. See [3,11,15] for a more general formulation using low-rank Gaussians and mixture of Gaussians.
If my understanding is correct, the authors consider a special case with $\gamma =0$ and differentiable $f(z)$ using an isotropic Gaussian approximation $\Sigma=\lambda I$. The authors should explain to readers why using an isotropic covariance instead of a diagonal covariance $\Sigma$.  Note that an isotropic covariance is a special case of a diagonal covariance.
Thus, many theoretical and empirical results in evolution strategies and variational inference can be readily applied to the isotropic setting considered in this work.

* Why do the authors use gradient descent instead of natural gradient descent in Algorithm 2? It is well known in variational inference [2,3,5,8,9,12], variational optimization [8,17] and evolution strategies [1,6,7] that natural gradient descent outperforms gradient descent in the Gaussian continuation.

* Existing works such as [13] in evolution strategies also consider convergence analysis. The authors should discuss these related works. The authors should discuss if assumptions 1-2 are weaker than assumptions used in the existing works in evolution strategies .

* Eq (6) is not new. It is known as Price’s Theorem or Stein’s Lemma or the heat equation identity. More general results such as gradient identities for a full covariance can be found at [4].

* If my understanding is correct, Eq (8) is the REINFORCE gradient estimator (see [4]) for $\lambda$, where the zero-order information of function $f$ is used.  A better gradient estimation using the first-order or second-order information of function $f$ has been considered in variational inference (see [3,4,7]). The gradient estimation can be used to estimate a diagonal or full covariance by using only one MC sample.


* The authors should explicitly mention how to estimate/compute the gradient with respect to $\theta$ in Eq (7). If my understanding is correct, the authors use the reparameterization gradient for the mean $\theta$ (see Bonnet’s Theorem in [4]), where the authors compute the gradient with respect to $\theta$  by using the first-order information of function $f$. Is my understanding correct? If so, this estimation is well-known.

* Why do the authors use the max/clipping operation in Algorithm 2 to handle the positivity constraint of $\lambda$?
A better approach is to reparametrize $\lambda$ as $\lambda=\exp(\eta)$ and learn the unconstrained parameter $\eta$ instead.
See [2,5,14] for handling the positive-definite constraint in covariance $\Sigma$.

* The regularization in Sec 2.5 is known as using a Gaussian prior in variational inference. This approach has been considered in [2,3,8,9]. The authors should discuss whether the regularization leads to a non-zero $\lambda$ at the optimal without the max/clipping operation in Algorithm 2.  The regularization may lead to a zero solution of $\lambda$ only when this clipping operation is used. This regularization may not work well if the max/clipping operation in Algorithm 2 is disabled. It is hacky to perform gradient decent in a constrained set (e.g., $\lambda>0$) since the authors have to use the clipping operation to satisfy the constraint.

* The saddle point problem in Sec 2.4 has been considered in evolution strategies (see [10]). The authors should discuss this related work.


* The numerical results are very weak.  For example, the MNIST experiment is weak. See [7,17,19] for strong experiments.
No baseline is included. Existing approaches such as [1,2,3,6,7] should be considered as baselines.
The method of [1] can also be easily adapted to learn a Gaussian convolution by using the gradient identities considered in [4].
Other approaches such as [2,3,14] also jointly estimate a diagonal covariance $\Sigma=\mathrm{diag}(\lambda)$ and the mean $\theta$ using natural gradient descent. Recall that an isotropic covariance is just a special case of a diagonal covariance.

References:

[1] Wierstra, et al. "Natural evolution strategies." The Journal of Machine Learning Research 15.1 (2014): 949-980.

[2] Lin et al.  "Handling the positive-definite constraint in the Bayesian learning rule." International conference on machine learning. PMLR, 2020.

[3] Lin et al. "Tractable structured natural-gradient descent using local parameterizations." International Conference on Machine Learning. PMLR, 2021.

[4] Lin et al.   "Stein's Lemma for the Reparameterization Trick with Exponential Family Mixtures." arXiv preprint arXiv:1910.13398 (2019).

[5] Salimbeni et al. "Natural gradients in practice: Non-conjugate variational inference in Gaussian process models." International Conference on Artificial Intelligence and Statistics. PMLR, 2018.

[6] Hansen, Nikolaus. "The CMA evolution strategy: a comparing review." Towards a new evolutionary computation: Advances in the estimation of distribution algorithms (2006): 75-102.

[7]  Lyu, et al. "Black-box optimizer with stochastic implicit natural gradient." Machine Learning and Knowledge Discovery in Databases. Research Track: European Conference, ECML PKDD 2021, Bilbao, Spain, September 13–17, 2021, Proceedings, Part III 21. Springer International Publishing, 2021.

[8] Khan, et al. "Fast and scalable bayesian deep learning by weight-perturbation in adam." International conference on machine learning. PMLR, 2018.

[9] Khan, et al.  "Conjugate-computation variational inference: Converting variational inference in non-conjugate models to inferences in conjugate models." Artificial Intelligence and Statistics. PMLR, 2017.

[10] Glasmachers, Tobias. "The (1+ 1)-ES Reliably Overcomes Saddle Points." International Conference on Parallel Problem Solving from Nature. Cham: Springer International Publishing, 2022.

[11] Khan, et al.  "The Bayesian learning rule." arXiv preprint arXiv:2107.04562 (2021).

[12] Hoffman, Matthew D., et al. "Stochastic variational inference." Journal of Machine Learning Research (2013).

[13] Glasmachers, Tobias. "Global convergence of the (1+ 1) evolution strategy to a critical point." Evolutionary computation 28.1 (2020): 27-53.

[14] Glasmachers, Tobias, et al. "Exponential natural evolution strategies." Proceedings of the 12th annual conference on Genetic and evolutionary computation. 2010.

[15] Lin et al. "Fast and simple natural-gradient variational inference with mixture of exponential-family approximations." International Conference on Machine Learning. PMLR, 2019.

[16] Möllenhoff, Thomas, and Mohammad Emtiyaz Khan. "SAM as an Optimal Relaxation of Bayes." arXiv preprint arXiv:2210.01620 (2022).

[17] Khan, et al. "Variational adaptive-Newton method for explorative learning." arXiv preprint arXiv:1711.05560 (2017).

[18] Staines, Joe, and David Barber. "Variational optimization." arXiv preprint arXiv:1212.4507 (2012).

[19] Salimans, Tim, et al. "Evolution strategies as a scalable alternative to reinforcement learning." arXiv preprint arXiv:1703.03864 (2017).

**Questions:**

See the weakness section.

---

### Official Review · Reviewer_jwZM · 2023-11-02

**Soundness:** 3 good
**Presentation:** 3 good
**Contribution:** 1 poor
**Rating:** 3
**Confidence:** 4

**Summary:**

Motivated by deep learning applications, the authors consider the problem of non-convex optimization. Following up on recent works on noise injection, the authors explore the idea of minimizing a sequence of functions that approximate the original problem. Assuming there exists a continuous path of minimizers for any value of the hyper-parameter $\lambda$, the authors prove that it is sufficient to minimize the functions for different values of $\lambda$ as long as they don't change too quickly. The authors also give examples of challenging situations where the continuation approach might not work well, and propose to use regularization to tackle those cases and use reparameterization $\lambda=\exp(\theta)$ to enforce non-negativity of $\lambda$.

**Strengths:**

1. The topic seems promising as noise injection has been previously shown to improve the convergence of SGD in certain cases.
2. The proposed approach is tested on problems somewhat relevant to deep learning.

**Weaknesses:**

Unfortunately, the theoretical contribution itself is quite limited. The results given in Theorems 1 and 2 are straightforward to obtain, and, as far as I can see, there is no theoretical result concerning the impact of Monte Carlo sampling or regularization.

My other major concern is about the utility of the approach in deep learning. As far as I can see, it is very likely to fail help when working with neural networks due to their within-layer permutation invariance. In particular, I suspect that 0 would always be a critical point for the loss function despite being sub-optimal. For example, take function $f(\theta_1, \theta_2) = (\theta_1 \theta_2 - 1)^2$. Then, it easy to show that with Gaussian kernel $g(\theta, \lambda) = (\theta_1 \theta_2 - 1)^2 + \lambda\theta_1^2 + \lambda\theta_2^2$. It is trivial to see that $(\theta_1, \theta_2) = (0, 0)$ is a critical point for any value of $\lambda$. Therefore, for large $\lambda$, we will converge to 0, and then we will stay there forever. This can be generalized to an arbitrary number of variables. I also think that due to the symmetry of neural networks, the most likely outcome is that with large $\lambda$, the solution would be all zeros. Since this point is also a critical point for most neural networks (due to permutation invariance), we will get stuck there. The authors discuss a similar issue in Section 2.4, but I do not how Monte-Carlo estimates or regularization from Section 2.5 would help.

Minor weaknesses:
1. The experiments are performed on small-scale problems such as MNIST, 2D problems. In my opinion, it could be ok if the paper had a good theoretical contribution.
2. The text refers to the Appendix a bit too much. It makes it harder to read the paper since the reader has to jump back and forth between the main body and the appendix.
3. The initial continuation parameter seems to require tuning, which suggests the approach is not very easy to use in practice.

**Questions:**

1. Is it possible to make the approach provably useful for objectives such as $f(\theta_1, \theta_2) = (\theta_1 \theta_2 - 1)^2$?
2. Can the authors provide formal guarantees assuming we minimize $g(\theta, \lambda)$ using SGD?